# Physiological Demands of Extreme Obstacle Course Racing: A Case Study

**DOI:** 10.3390/ijerph16162879

**Published:** 2019-08-19

**Authors:** Timothy Baghurst, Steven L. Prewitt, Tyler Tapps

**Affiliations:** 1College of Education, Florida State University, Tallahassee, FL 32306, USA; 2Department of Health and Human Performance, Texas A&M University-Commerce, Commerce, TX 75428, USA; 3School of Health Science & Wellness, Northwest Missouri State University, Maryville, MO 64468, USA

**Keywords:** accelerometer, endurance training, energy expenditure, intensity, heart rate

## Abstract

Obstacle course races are a popular source of recreation in the United States, providing additional challenges over traditional endurance events. Despite their popularity, very little is known about the physiological or cognitive demands of obstacle course races compared to traditional road races. The purpose of this study was to investigate the physiological effects of participation in an extreme obstacle course race. The participant was a 38-year-old Caucasian male, who completed an extreme obstacle course race over a 24-h period. Exercise intensity, steps taken, energy expenditure, and heart rate were recorded over the event’s duration using an Actigraph Link GT9X-BT accelerometer and a Polar heart rate monitor. Results reflected the unique nature of obstacle course racing when compared to traditional endurance events, with ups-and-downs recorded in each variable due to the encountering of obstacles. This case study provides a glimpse into the physiological demands of obstacle course racing, and suggests that the cognitive demands placed on competitors may differ to traditional endurance events, due to the challenges of obstacles interspersed throughout the event. With the popularity of obstacle course racing, and to enhance training opportunities, improve performance, and decrease the incidence of injuries, future research should further investigate the physiological and cognitive demands of obstacle course races of various distances and among diverse populations.

## 1. Introduction

According to Running USA [1], there were just over 18 million road race participants in 2018, which is a number that has slowly but steadily declined from a record 19 million participants in 2013. This gradual reduction in participation may be in part due to non-traditional events [1], such as obstacle course racing (OCR) which emerged in 2009, and has grown quickly. The development of OCR has brought a new competitive experience to running, as it requires participants to complete challenging obstacles that include jumping, climbing, and crawling, over a variety of rough terrains and environments including hills, water, and mud. Just as with traditional road races, OCR provides events for athletes of varying levels, with distances from three miles to multi-day competitions covering upwards of 100 miles [2]. Events for children have also emerged as popular [3,4].

The number of participants in OCR events by year is unclear; however, Nikolova [5] estimated approximately 500,000 annually in the United States. The increasing popularity of these events over the past several years has led to calls to add OCRs to the Olympic sports [6], and even physical education curricula [7]. Distances for OCR events vary, and while most participants engage in short races of less than five miles, approximately 50,000 participants engage in events longer than 10 miles. In contrast to road running, participation in OCR is 64% male; increasing to 69% male when the event exceeds 10 miles in length [5].

The concept of using obstacle courses for training dates to the Roman Empire, when it was used for military preparation [8]. As an integral part of modern-day military training, obstacle courses are introduced in the first phase of basic combat training (BCT) and are present throughout the duration of training [9]. In addition, obstacle courses are frequently used as a part of physical education curricula [10,11].

Today, obstacle courses have been integrated into racing formats, and the physical and mental demands placed upon participants can be significant [12]. Rabb and Coleby [12] reported that medical personnel at OCR events should anticipate up to a 5% injury rate, with 4.5% requiring emergency department treatment. Such injuries occur due to environmental (e.g., weather, type of obstacle, apparel malfunctions) and physical conditions (e.g., hydration, mental/physical fatigue).

Despite their popularity, little data exists on the physiological or cognitive demands of OCR, and particularly on those events that might be considered extreme in length. However, parallels to ultra-endurance events are possible. Ultra-endurance events are characterized by a steady intensity throughout the first portion of the race, before a mid-to-late race fatigue that causes a decrease in intensity. Then, as the end of the race nears, there is a marked increase in intensity, likely representing a final push to the finish [13]. While the participants in ultra-endurance road races experienced relatively linear changes in intensity throughout the event, intensity did not remain constant throughout the race. Similar findings were revealed using heart rate as a measure of intensity among female runners. However, greater variability was discovered in heart rate response among female off-road racers (cross-country, fell runners, and orienteering) when compared to road racers [14]. Researchers attributed this variability to runners adapting to the changes in terrain and environment, when compared to a road race environment [14].

There are plenty of data to highlight the physiological demands of traditional endurance events. However, despite their popularity [15], little research has investigated how OCR, with its additional obstacle challenges such as swimming, climbing, crawling, and balancing [16], might affect the physiological demands of these athletes. Therefore, the purpose of this study was to investigate the physiological effects of participation in an extreme obstacle course race.

## 2. Materials and Methods

### 2.1. Participant

The participant was a 38-year-old Caucasian male, whose height was 183 cm (6′0”) and mass was 90.7 kg (200 lbs). He was an experienced racer in a variety of running and obstacle events but participated in his first extreme OCR in the present study. To qualify, he finished in the top 10% at a regional OCR. To prepare for the event, the participant completed a 50 km running plan with additional strength training exercises including push-ups, pull-ups, kettle bell, and grip strength exercises. This training plan was designed by the participant’s OCR coach.

### 2.2. Instruments

An Actigraph Link GT9X-BT accelerometer (ActiGraph, LLC, Pensacola, FL, USA. Polar OY, Kempele, Finland) was worn on the participant’s non-dominant (left) wrist for the duration of the event. The participant also wore a Polar heart rate monitor around the chest that was synced with the accelerometer. Published race results were also obtained from the event’s website to add depth to the physiological measures.

The extreme OCR was held in November in Las Vegas, Nevada, and was comprised of a five-mile course with 21 obstacles. The race began with a seven-mile run, after which the obstacles were opened for the remaining length of the event. Participants faced up to an additional nine penalty obstacles on each lap, if they elected to skip or could not complete an obstacle on the course. Therefore, some laps took longer based on the success or failure at each obstacle, rather than due to participant fatigue. Participants had 24 h to complete as many laps of the course as possible. If participants were in the middle of a lap at the end of the 24 h period, they were afforded an additional 2 h to complete that lap.

The course included a variety of water obstacles, and participants could encounter up to 13 opportunities to navigate through water on each lap, which varied depending on the number of obstacles added due to penalties. Elevation also created an added challenge, as each lap presented an 800 foot differential. The course was primarily grass and dirt, and challenges comprised of activities requiring climbing, crawling, lifting, and pushing. Some activities also focused on mental and balance skills, such as maneuvering a metal implement through electrified wires. Temperature ranged throughout the 24 h period, but there was no rain.

At the conclusion of each lap, participants were allowed to rest and refuel for as long or as short a duration as desired. Rules allowed for a maximum of two support personnel to assist the participant, and during this period, participants were permitted to change clothes.

### 2.3. Procedure

A university Institutional Review Board (IRB) approved this study as pre-existing data not originally intended for publication (ED-19-94). Height, mass, date of birth, handedness, and ethnicity were entered into the Actigraph ActiLife 6 software (Actigraph, Pensacola, FL, USA) to calibrate the accelerometer. The Actigraph was set to record data at 10 s epochs. Data collection began one hour prior and was completed one hour following the event. The participant wore a heart rate monitor around his chest for the duration of this time; however, heart rate recordings were intermittently interrupted because of obstacles encountered such as water. After the race was completed, accelerometer data were downloaded using ActiLife 6 software, converted to 60 s epochs, and analyzed using Freedson [17] algorithms.

## 3. Results

Heart rate (bpm), intensity levels (metabolic equivalents (METS)), steps taken, and energy utilization (kcals) were all measured using the Actigraph accelerometer and Polar heart rate monitor (Actigraph, Pensacola, FL, USA). Data were calculated using Actilife software, which accepts Polar heart rate data. The participant began the race at 15:00 and ended the race at 15:58 the following day, where the race time total was 24:58. During this period, 55 miles and 11 laps were completed. The participant finished 28th of 207 in his age group, 116 of 988 by sex, and 125 of 1140 overall.

The participant’s race pace trended toward an overall increase in time per lap over the course of the event (Table 1). During the first lap, the participant recorded a sub 10 min-mile pace, while during the last lap the participant recorded a mile pace of over 35 min, with an average 27.25 min-mile throughout the race.

Prior to beginning the race, the participant’s heart rate was 71 bpm, which reached a maximum heart rate of 174 bpm during the extreme OCR. Using an age predicted maximum heart rate equation (HR_max_ = 220 − age), which matched the participant’s self-reported HR, he was working at just over 95% of his HR max [18]. The participant’s average heart rate, and a graphical representation of his heart rate, were not reported due to inconsistent heart rate readings that were attributed to interference from the water obstacles.

Intensity was measured using METS (metabolic equivalent), a measure of relative intensity [18]. Average intensity for the race was recorded at 3.4 METS. Peak intensity was reached 1 h into the race (4.8 METS), and the lowest recorded intensity was at 22 h (2.3 METS). Figure 1 demonstrates the slow decline in METS from the first few hours, before more dramatic peaks and troughs as the race neared completion.

Participant exercise intensity was calculated using the Freedson [17] algorithm, and physical activity cut-points. The majority of the time (58.54%) was spent in moderate activity, and the remainder was spent in light (38.14%) physical activity, or as sedentary (3.33%).

Steps per hour are presented in Figure 2. Like peak intensity, peak steps per hour were reached in the first hour of the race at 9081 steps per hour. The lowest steps per hour occurred 22 h into the race, where 1977 steps were taken during that hour. In total, 79,129 steps were taken over the duration of the race, with an average of 2874 steps per hour.

Energy utilization from an hour prior and an hour following the race is shown in Figure 3. Like other variables, a steep increase in energy utilization occurred during the first hour of the race, where peak kcals (435) were achieved. Low energy utilization occurred at hour 22, where only 90 kcals were burned. Over the course of the extreme OCR, 6838 kcals, or 244 kcals per hour, were utilized.

## 4. Discussion

The purpose of this study was to investigate the physiological effects of participation in an extreme obstacle course race (OCR). The peak measurements were recorded one hour into the race, with lows occurring at 22 h in each accelerometer measure recorded: Heart rate, steps, exercise intensity, and energy utilization. The peaks coincided with the completion of a multi-mile run before the obstacles were opened. The peak intensity, steps, and kcals also occurred during this stage of the event.

Although the heart rate readings were inconsistent, which was possibly due to the movement of the heart rate monitor as the participant crawled and swam, a range of 103 bpm (71–174 bpm) was recorded. The participant’s peak heart rate reflected almost 96% of the age-predicted maximum heart rate, which is classified as correlating to vigorous or high intensity exercise [18]. This finding is interesting to consider when compared to the METS recorded, which indicates that at no point did the participant reach the classification of vigorous exercise. The intensity remained relatively stable from 19:00 to 01:00, which was between 4 h–10 h into the race, and then greater variation in exercise intensity became apparent after the 12 h mark.

Exercise intensity measurements indicate that the participant was classified as participating in moderate activity (*M* = 3.4 METS) during the majority or the race, falling below moderate activity at 10:00 and 13:00, hours 16 and 19 respectively, outside of the start and conclusion of the race. The up-and-down nature of the participant’s exercise intensity in the last half of the race likely represents the varying intensity provided by the obstacles, combined with possible fatigue and the utilization of different energy systems.

Freedson [17] cut-points (see Freedson’s article for the specific cut-point and algorithms used) reflect similar findings, suggesting that the participant participated in moderate exercise for just over 58% of the time, with approximately 38% of the extreme OCR considered as light exercise, and just over 3% recorded as sedentary activity. It is likely that the sedentary classifications occurred when the participant changed clothes or rested between laps.

A peak heart rate reading of 174 bpm suggested that the participant reached vigorous intensity exercise at one point, although the MET calculations and Freedson [17] cut-points did not reflect the same data. Further investigation revealed that the peak heart rate coincided with the highest intensity measured in the first hour of the race, which would suggest a correlation of the two measures

Reaching a level of over 79,000 steps over an almost 25 h period far exceeds a typical daily goal of 10,000 steps [19]. Such excessive physical activity may lead to overuse injuries [20]. In ultra-endurance events, overuse injuries may arise from overtraining or from injury during the event [12,19,21].

Exercise intensity oscillated throughout most of the race, with a relatively steady decline in mile pace as time progressed. The up-and-down nature of the exercise intensity appears to be unique to off-road running [14]; however, the pace results were similar to those of an ultra-marathon runner, with the relatively steady decline of the mile pace throughout the race [14]. The fast pace in the first lap could be attributed to inexperience, as this was the participant’s first extreme event. However, it should also be noted that the event began with a seven-mile run, which may also account for the much higher levels of physiological data.

## 5. Limitations and Future Research

The exploratory nature of this case study provides for multiple limitations and opportunities for future research. Baseline measurements for variables, such as resting heart rate and VO^2^_max_, would provide for a more in-depth look at the relative intensities and changes in intensity over time. Similarly, assessing the basal metabolic rate (BMR) would provide insight into the understanding of extreme OCR participation on kilocalorie expenditure. With the nature of OCR and its high levels of energy expenditure, it would be beneficial to understand the pre-, during-, and post-competition nutrition practices of current OCR participants. Similar research has been conducted with ultramarathon runners [21], but a gap in the literature remains with OCR.

The inconsistent measures that were provided by the heart rate monitor proved to be a limitation of the study, as the minimum and maximum heart rates may have occurred during water obstacles, where the monitor malfunctioned. Future studies should utilize a heart rate monitor that is able to function with the presence of water. Additionally, the field nature of such competitions provides limitations to measurements when compared with laboratory measurements in a controlled setting. The energy expenditure measured using accelerometer software may also have its own limitations. Future research should also include multiple participants, in order to understand the different physiological responses to extreme OCR, as well as by attempting to measure data in as controlled an environment as possible while outside of a lab setting. Changes in temperature and weather conditions were not measured, and it would also be interesting to understand how accelerometers might be used to influence or regulate pace in such events. Research suggests that accelerometers may not be of value unless used with specific goals in mind [4]. Therefore, evaluating the relative benefits of participants being provided with physiological data in a real-time environment should be investigated. Lastly, no psychological measures were undertaken. Participants often undergo both physical and environmental challenges, including: Heat and cold exposure, dehydration, nutritional deficiencies, and sleep deprivation, that could lead to a variety of psychological challenges.

## 6. Conclusions

Obstacle courses hold a long tradition in society that began with the Roman Empire, and that remain today in physical education, military training, and more recently among recreational and professional athletes. The evolution of the running industry that now incorporates OCR, has changed the running landscape. Thousands of athletes and weekend warriors participate in OCR each year, but despite the large numbers of participants, little is known of the demands and physiological effects of OCR. This case study has given a glimpse into the physiological responses to OCR through understanding the calories expended, the intensity of the exercise, and the steps taken throughout the course of the race. Findings highlight the need for future research to investigate how participants can be better prepared mentally and physically, to help prevent injuries, and to improve the overall experience of OCR.

## Figures and Tables

**Figure 1 ijerph-16-02879-f001:**
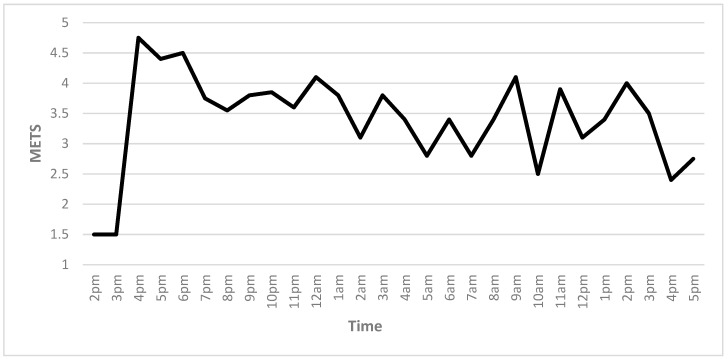
Participant intensity in metabolic equivalents (METS) over time (<3 = Low METS; >6 = High METS).

**Figure 2 ijerph-16-02879-f002:**
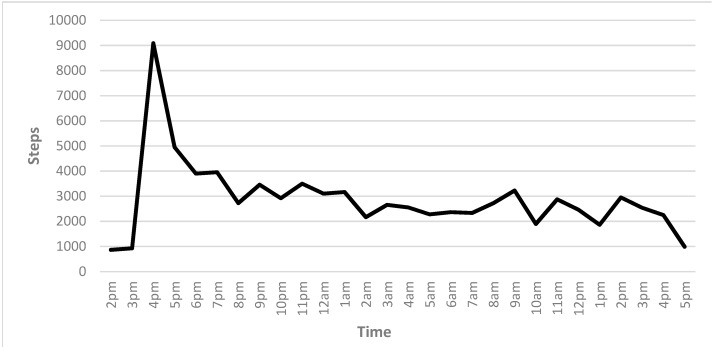
Recorded steps as measured by Actigraph.

**Figure 3 ijerph-16-02879-f003:**
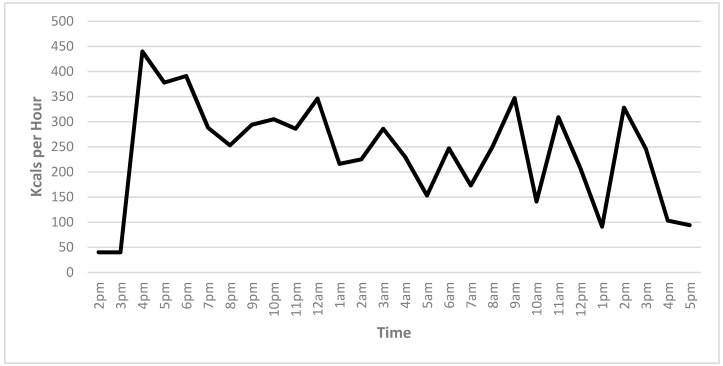
Rate of energy utilization over time.

**Table 1 ijerph-16-02879-t001:** Participant splits and pace per lap over the duration of the obstacle course race (OCR).

Lap Number	Split Time	Pace Per Mile
1	45:07	9.02
2	1:02:07	12.26
3	1:39:56	20.00
4	2:03:13	24.39
5	1:49:11	21.51
6	1:58:13	23.39
7	2:21:11	28.15
8	2:20:56	28.12
9	2:03:23	24.41
10	2:22:18	28.28
11	2:59:13	35.51

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
