# Peer review of "Physiological Demands of Extreme Obstacle Course Racing: A Case Study"

_ijerph, 2019, doi:10.3390/ijerph16162879_

Round 1

Reviewer 1 Report

Well written and an interesting study. 

Author Response

Thank you for your comments and kind words. The suggested revisions were all grammatical and formatting in nature. We have accepted all suggestions and have incorporated them into the manuscript as recommended.

Reviewer 2 Report

General Comments

Some additional information about the course would be valuable, such as more information about the obstacles and time to complete each obstacle, environmental conditions, running surface. 

Some details about the methods for determination of METS and energy utilization should be provided.

Of course, the obvious significant limitation includes that this is an observational study of a single case. 

Specific Comments

Lines 30-31.  This second sentence makes is sound as though the participant data for road racing are inclusive of obstacle course racing.  Is that the case?

Lines 53 and 65.  The word “data” is the pleural form of “datum” so it should be “little data exist….” and “There are plenty of data….”

Lines 109 and 115.  The first sentence of each of these paragraphs is duplicative of the methods. 

Line 109.  How did the participant place overall and within his age group?  What was the field size?

Line 112.  Please indicate that you are referring to time per lap. 

Line 118.  If heart rate data are suspect, then how do you know that you have an accurate peak value?

Line 133 and elsewhere.  It should be made clear that you are reporting the lowest steps per hour for the full hour beginning at 22 hours into the race.  Similar clarifications are necessary elsewhere. 

Figure 2.  The low point at 1pm doesn’t seem to be at 1677. 

Figure 3.  The title should refer to rate of energy utilization, and the y-axis should be Kcals per hour. 

Line 148.  I thought you didn’t know the lowest heart rate, but here you refer to it being during the 22-23 hour. 

Line 174.  There is a well known relationship between heart rate and oxygen uptake (energy expenditure) so I’ve confused by this statement.

Line 174.  Are you sure reference 19 is correct?

Line 204.  Should be undertaken. 

Line 218.  Emilee Bounds is not listed as an author, and you do not indicate what contribution was made by Tyler Tapps. 

Author Response

Thank you for your thoughtful comments that have made great improvements to the quality of this article. We have explained how we have incorporated your revisions into the manuscript below. Again, our thanks!

Comments and Suggestions for Authors

General Comments

Some additional information about the course would be valuable, such as more information about the obstacles and time to complete each obstacle, environmental conditions, running surface. 

Additional information about these concerns has been added.

Some details about the methods for determination of METS and energy utilization should be provided.

More details have been provided, although the data were computed using the Actilife software. How this is actually computer (i.e., the mathematical formula used) is not shared by Actigraph. Of course, this is a limitation.

Of course, the obvious significant limitation includes that this is an observational study of a single case. 

But we think the data is valuable and a start for more research in this area.

Specific Comments

Lines 30-31.  This second sentence makes is sound as though the participant data for road racing are inclusive of obstacle course racing.  Is that the case?

Yes, OCR is included in this data.

Lines 53 and 65.  The word “data” is the pleural form of “datum” so it should be “little data exist….” and “There are plenty of data….”

Great catches thank you!

Lines 109 and 115.  The first sentence of each of these paragraphs is duplicative of the methods. 

These sentences have been modified.

Line 109.  How did the participant place overall and within his age group?  What was the field size?

Line 112.  Please indicate that you are referring to time per lap. 

Modified as suggested.

Line 118.  If heart rate data are suspect, then how do you know that you have an accurate peak value?

It is an assumption, of course, but the interruptions in the water-events are unlikely to have affected peak value. It is possible. However, the interruptions were not for a large portion of the event, but enough to affect a graph, which is why it was reported.

Figure 2.  The low point at 1pm doesn’t seem to be at 1677. 

Thank you for the catch. The 6 should have been a 9.

Figure 3.  The title should refer to rate of energy utilization, and the y-axis should be Kcals per hour. 

Changed as suggested.

Line 148.  I thought you didn’t know the lowest heart rate, but here you refer to it being during the 22-23 hour. 

Within the constraints outlined we reported what we could, recognizing that it might not have been a “true” value given the lack of HR reported during some water challenges. This has been added in the limitations.

Line 174.  There is a well known relationship between heart rate and oxygen uptake (energy expenditure) so I’ve confused by this statement.

This statement has been removed.

Line 174.  Are you sure reference 19 is correct?

Another good catch thank you.

Line 204.  Should be undertaken. 

Corrected thank you.

Line 218.  Emilee Bounds is not listed as an author, and you do not indicate what contribution was made by Tyler Tapps. 

Corrected thank you.

Reviewer 3 Report

Physiological Demands of Extreme Obstacle Course Racing: A Case Study

The topic is of interest to both physiologists and practitioners, giving the increasing interest in this type of competitions. The study is of a descriptive type, as few have looked at Obstacle Course Racing (OCR) before. This case study is nicely put together and offer an enlightening description of OCR that is useful for developing further studies on the issue.

I have some comments to the text. In the Introduction, an “end spurt” is named a characteristic of ultra-endurance events, but the figures depicting heart rate and energy consumption does not show such a feature, so I hope You can make some comments on that, is it the obstacles that prevent an end spurt?

Energy expenditure is not measured here, but estimated using accelerometer and heart rate data. You should therefore also in the discussion make comments on how accurate such an estimation will be. There are several pitfalls when doing such an analysis.

To better present the energy expenditure, Figure 1 could have horizontal lines indicating low and high values (low < 3 MET, high > 6 MET). Also, You could combine Figure 1 and 3 to highlight the correlation between HR and MET.

Line 65: Can You indicate on the figures when the rest periods were taken?

You have pre-race body mass, what about post-race? Use rather “body mass” than “weight” as mass is what You have been measuring.

Line 82-83: ActiGraph, LLC, Pensacola, FL, USA. Polar OY, Kempele, Finland

Line 86: five-mile (8 km) (?).

The Freedson [17] algorithm and physical activity cut points should be briefly described to the reader. Also, I would like to see a comparison of their values to others, see Trost et al. Med. Sci. Sports Exerc., Vol. 43, No. 7, pp. 1360–1368, 2011.

As this obviously is a person well used to training, his maximal heart rate is probably known to him, better to use that than the 220-age formulae.

Your comment in line 165 on different energy systems deserves a broader explanation, although fat metabolism is not measured by You.

Please use SI units, with imperial units in brackets (if needed).

Author Response

Comments and Suggestions for Authors

Physiological Demands of Extreme Obstacle Course Racing: A Case Study

The topic is of interest to both physiologists and practitioners, giving the increasing interest in this type of competitions. The study is of a descriptive type, as few have looked at Obstacle Course Racing (OCR) before. This case study is nicely put together and offer an enlightening description of OCR that is useful for developing further studies on the issue.

Thank you for your comments.

I have some comments to the text. In the Introduction, an “end spurt” is named a characteristic of ultra-endurance events, but the figures depicting heart rate and energy consumption does not show such a feature, so I hope You can make some comments on that, is it the obstacles that prevent an end spurt?

Good question. Although speculation, the lack of experience of this participant with such events may account for the lack of end spurt. Essentially, the participant may not have paced himself appropriately and therefore struggled to finish. The data would suggest such. It would be interesting to compare novice vs experienced ultra OCR runners in this respect.

Energy expenditure is not measured here, but estimated using accelerometer and heart rate data. You should therefore also in the discussion make comments on how accurate such an estimation will be. There are several pitfalls when doing such an analysis.

Agreed and we have added this as a limitation.

To better present the energy expenditure, Figure 1 could have horizontal lines indicating low and high values (low < 3 MET, high > 6 MET). Also, You could combine Figure 1 and 3 to highlight the correlation between HR and MET.

We have added in a description within the Figure title; however, we elected not to combine the graphs. Our hope is that our article reaches the general running population, and we believe that combining the two graphs would complicate the content.

Line 65: Can You indicate on the figures when the rest periods were taken?

Unfortunately not. We did not ask this of the participant. In his own words “I went straight to bed!”

You have pre-race body mass, what about post-race? Use rather “body mass” than “weight” as mass is what You have been measuring.

Body mass has been changed in the manuscript. It was not measured post event. Again, as a case study it was a unique opportunity, but provides a starting point for future studies.

Line 82-83: ActiGraph, LLC, Pensacola, FL, USA. Polar OY, Kempele, Finland

Added as suggested.

Line 86: five-mile (8 km) (?).

All units were in miles.

The Freedson [17] algorithm and physical activity cut points should be briefly described to the reader. Also, I would like to see a comparison of their values to others, see Trost et al. Med. Sci. Sports Exerc., Vol. 43, No. 7, pp. 1360–1368, 2011.

Respectfully, we disagree that this is of primary importance to the discussion. Our goal is to present case study data for reference to those interested in pursuing future research. We believe that a discourse on cut points and algorithms in the discussion section would detract from the overall study purpose. However, we have included in the manuscript to refer to the Freedson article for more information.

As this obviously is a person well used to training, his maximal heart rate is probably known to him, better to use that than the 220-age formulae.

It was the same value, which we have added to the manuscript.

Please use SI units, with imperial units in brackets (if needed).

This is something we think the journal editor should decide, as imperial units were part of the original article submission. We are happy to change if necessary.

Reviewer 4 Report

Authors should be congratulated. Their study is scientifically sound, well-written, easy to follow and a good contribution to the field. I am happy to advise to accept as it is, in its current form.

Author Response

Thank you very much for your kind words and supportive response.

Round 2

Reviewer 2 Report

Specific Comments

Lines 30-31. You have indicated that the Running USA road running data include obstacle course racing data. That seems odd, since obstacle course racing is not a road race. Also, the indicate that the road racing participation data you refer to are from 2017, yet the citation seems to be from 2015, and the provided webpage will not open for me. Line 73. You have changed “weight” to “mass”, but pounds is a unit of weight. Most journals want data reported in metric units, so the height should be reported in cm and the mass in kg. Line 103. No temperatures are provided.

Author Response

Thank you again for your comments and suggestions for further revision. We have addressed both issues as requested. Your time and thoughts on this manuscript have been appreciated.

Lines 30-31. You have indicated that the Running USA road running data include obstacle course racing data. That seems odd, since obstacle course racing is not a road race. Also, the indicate that the road racing participation data you refer to are from 2017, yet the citation seems to be from 2015, and the provided webpage will not open for me.

Thank you for catching this. We had updated the statistic and also provided an updated reference. We have revisited this statistic and contacted the organization directly to confirm. The data general does NOT include OCR events, which helps to explain why road race numbers appear to be declining. A new reference has been provided.

Line 73. You have changed “weight” to “mass”, but pounds is a unit of weight. Most journals want data reported in metric units, so the height should be reported in cm and the mass in kg. Line 103. No temperatures are provided.

We have included both cm/in and kg/lb to satisfy a global readership. We cannot provide temperature, as it was not recorded on the day and it ranged throughout the 24 hour period. This has been added as a limitation.